# PlasticEnz: An integrated database and screening tool combining homology and machine learning to identify plastic-degrading enzymes in meta-omics datasets

Anna Krzynowek, Jasper Snoeks, Karoline Faust*

Department of Microbiology, Immunology and Transplantation, Rega Institute for Medical Research, Laboratory of Molecular Bacteriology, KU Leuven, Leuven, Belgium

* karoline.faust@kuleuven.be

## Abstract

*PlasticEnz* is a new open-source tool for detecting plastic-degrading enzymes (plastizymes) in metagenomic data by combining sequence homology-based search with machine learning techniques. It integrates custom Hidden Markov Models, DIAMOND alignments, and polymer-specific classifiers trained on ProtBERT embeddings to identify candidate depolymerases from user-provided contigs, genomes, or protein sequences. PlasticEnz supports 11 plastic polymers with ML classifiers for PET and PHB, achieving F1 > 0.7 on an independent test set. Applied to plastic-exposed microcosms and field metagenomes, the tool recovered known PETases and PHBases, distinguished plastic-contaminated from pristine environments, and clustered predictions with validated reference enzymes. *PlasticEnz* is fast, scalable, and user-friendly, providing a robust framework for exploring microbial plastic degradation potential in complex communities.

## Author summary

Plastic pollution is a global problem, and one promising solution is to apply microbes to break them down. However, finding the enzymes responsible for this in complex environmental samples is not easy. We developed *PlasticEnz*, a free and easy-to-use tool that helps researchers identify plastic-degrading enzymes or "plastizymes" in metagenomic data. *PlasticEnz* combines traditional sequence similarity search methods with machine learning models trained on previously known plastizymes. It works with protein sequences, contigs, or genomes with ML components optimised for classification of two common plastizymes: PETases and PHBases. We tested *PlasticEnz* on both controlled lab experiments and real-world samples from plastic-polluted soils and clean environments. The tool successfully identified known plastic-degrading enzymes

**Data availability statement:** All relevant data underlying the findings of this study, including the raw data used for model training, benchmarking, intermediate files, scripts to generate figures and tables are provided within the manuscript, supporting information files, and Zenodo repository (10.5281/zeno-do.15395662). PlasticEnz is freely available at https://github.com/msysbio/PlasticEnz. All the raw data, training sets and scripts are available at 10.5281/zenodo.15395662.

**Funding:** AK recieved funding from Fonds Wetenschappelijk Onderzoek (FWO) PhD fellowship 1S03725N. https://fwo.be The funders had no role in study design, data collection and analysis, decision to publish, or preparation of the manuscript. AK recived salary from the funder.

**Competing interests:** The authors have declared that no competing interests exist.

and even helped distinguish between polluted and pristine sites. By making plastizyme detection more accessible, *PlasticEnz* enables researchers to better explore the microbial potential for plastic degradation, which could support future bioremediation efforts.

## Introduction

Plastic pollution is a growing environmental problem posing serious risks to ecosystems, wildlife, and human health [1–4]. Despite this, only a small fraction of the millions of tons of plastic waste generated each year is recycled or reused, with most ending up in natural environments [5,6]. The capacity of some microorganisms to biodegrade plastic polymers has attracted considerable attention as a potential strategy aimed at mitigating plastic pollution [7–14]. In particular, polyester plastics (e.g., polyethylene terephthalate (PET), polylactic acid (PLA), polycaprolactone (PCL)), which are widely used in textile and packaging manufacturing, are of interest for potential microbial bioremediation due to the presence of repeated ester bonds that can be targeted by various extracellular depolymerases [11,13,15,16]. Recent advances in bioinformatics combined with decreasing cost of whole-genome sequencing, have greatly improved our ability to screen complex communities for novel enzymes. However, existing computational tools often lack specificity for plastic substrates and involve complex multi-step pipelines that might not be accessible for researchers without bioinformatics expertise.

Several publicly available databases now catalogue protein sequences and associated metadata for enzymes involved in plastic degradation [17–19]. Commonly used approaches to identify candidate plastic-degrading enzymes involve large-scale homology searches against these databases using fast sequence aligners such as DIAMOND [20], Bowtie2 [21], or Minimap2 [22]. Other methods rely on domain-based annotation, such as screening for specific functional domains using tools like InterProScan [23] or Pfam [24]. In addition, custom Hidden Markov Motifs (HMMs) [25,26] built from curated enzyme sequences and tailored to specific plastic polymers have been employed before to improve search specificity [27,28]. More recently, the application of machine learning (ML) to protein function prediction is being increasingly explored, enabling the discovery of novel enzyme candidates that lack clear homology to known proteins [29–35]. In this context, ML models have been applied to predict plastic-degrading enzymes, as demonstrated by Jiang et al. [36,37]. However, these models were not tested on real-world metagenomics datasets and are not readily available to be applied to the user's own data.

To address these limitations, we developed *PlasticEnz*, an open-access tool that combines homology-based search using custom HMMs with machine learning prediction to improve the detection of plastic-degrading enzymes. In this study, we describe the development, testing, and application of *PlasticEnz*. The tool accepts protein sequences, genomic assemblies, or contigs as input, and identifies candidate plastic-degrading enzymes using a combination of custom HMMs, DIAMOND-based

homology searches, and optional machine learning classification. Specifically, HMM-based screening is available for poly(3-Hydroxypropionate (P3HP), poly(butylene adipate-co-terephthalate) (PBAT), polybutylene succinate (PBS), poly(butylene succinate-co-adipate) (PBSA), polycaprolactone (PCL), poly(ethylene adipate) (PEA), polyethylene tere-phthalate (PET), polyhydroxybutyrate (PHB), poly(3-hydroxybutyrate-co-3-hydroxyvalerate) (PHBV), and polylactic acid (PLA); DIAMOND-based searches are implemented for PBS, PBSA, PCL, PES [polyethersulfone], PHBV, and PLA; and machine learning predictions using XGBoost (default) and a more sensitive neural network model are currently available for PET and PHB (S1 Table). This integrated approach allows *PlasticEnz* to flexibly detect enzyme homologs across a wide range of plastic polymers with adjustable sensitivity and specificity. To facilitate accessibility, *PlasticEnz* is imple-mented as a command-line tool with streamlined output formats that include HMMER/DIAMOND outputs such as bitscore and E-values, ML prediction scores and normalized gene abundances (TPM/RPKM), making the results easier to interpret and compare across diverse metagenomic datasets.

We applied *PlasticEnz* to both controlled microcosm experiments and diverse field metagenomes, demonstrating its ability to discriminate plastic-exposed from pristine environments. In the Laguna Madre microcosm, PlasticEnz identified a strong enrichment of PHB depolymerase homologs in PHA biofilms, while PETase signals remained consistently low across all treatments, aligning with the findings of the original study. In plastic-contaminated soil samples, those collected in Sewapura and Varamin exhibited the highest abundance and prediction scores for both PET and PHB depolymerases, whereas pristine Kamchatka acid hot springs communities yielded negligible hits. Benchmarking against our test set, the ML classifiers achieved F1 values above 0.7 for PET and PHB, with XGBoost maximizing precision and the neural net-work showing enhanced sensitivity. Sequence-level analyses further confirmed that predicted homologs clustered closely with experimentally validated reference enzymes.

*PlasticEnz* offers a streamlined and accessible solution for identifying plastic-degrading enzymes in metagenomic data by combining homology-based and machine learning approaches. By integrating curated reference data with predictive models in a single pipeline, it enables researchers regardless of computational background to explore microbial plastic degradation potential across a wide range of environments.

## Results

### *PlasticEnz* database

The *PlasticEnz* database contains 213 unique protein sequences associated with plastic polymer degradation path-ways, extracted from 176 peer-reviewed studies. Each database entry includes detailed annotation of the enzyme's gene name, location (e.g., extracellular or cell-bound), enzyme classification, operon or gene cluster, targeted bond types, and catalytic domains. Enzymes are also linked to their associated reactions (substrate-product) and source organisms, for which marker genes or whole-genome sequences are provided. Furthermore, the database integrates cross-references to external databases (UniProt, NCBI). The comparison with other available *Plastizyme* databases is shown in Table 1.

**Table 1. Comparison of stored information across available plastic-degrading enzyme databases.**

| Feature | PlasticEnz | PlasticDB [18] | PAZy [19] |
|---|---|---|---|
| Reactions annotated | Yes (substrate–product pairs) | No | Limited (some polymer types) |
| Microorganism sequences | Genome or 16S rRNA sequence provided | No | No |
| Links to other databases | NCBI, UniProt | None | GenBank, Uniprot, PDB |
| Enzyme location (cellular) | Specified | Not available | Specified for some entries |
| Enzyme isolation environment | Specified for some enzymes | Not available | Not available |

## ML model evaluation

To assess the predictive performance of PlasticEnz, we compared three machine learning models (Neural Network, Random Forest, and XGBoost) for their ability to classify plastic-degrading enzymes across polymer substrates. This comparison aimed to determine which model best balances sensitivity and precision when identifying plastizyme candidates in metagenomic data (**Fig 1**). Overall, all models struggled to accurately classify PLA, PBAT, PBSA, PCL and PHA polymers. However, for PET and PHB, both the Neural Network and XGBoost models achieved F1 values above 0.7 (**Table 2**). The XGboost model demonstrated higher precision (0.95 and 1 for PET and PHB) than the Neural network (0.84 and 0.63 for PET and PHB), indicating a lower rate of false positive classifications (paired t-test, p-value < 0.01, t = -71.0). However, it

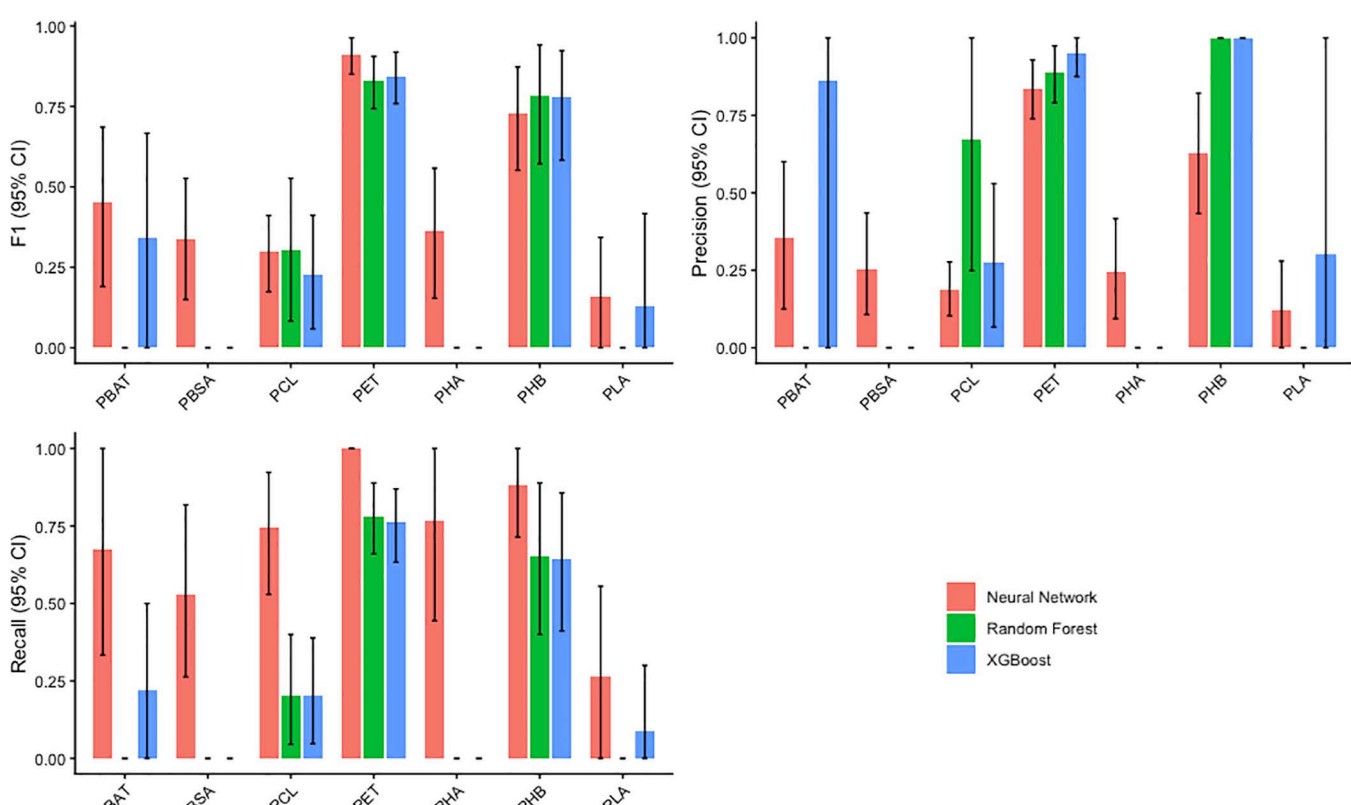

**Fig 1. Bar charts depicting the mean bootstrapped (n = 1000) evaluation metrics (F1, Precision, and Recall) for classification of each plastic-degrading enzyme class.**

**Table 2. Performance metrics of Neural Network and XGBoost models for PET and PHB classification. Numbers in the brackets represent 95% confidence intervals.**

| Model | Polymer | Precision | Recall | F1 |
|---|---|---|---|---|
| Neural Network | PET | 0.84 (0.74–0.93) | 1.00 (1.00–1.00) | 0.91 (0.85–0.96) |
| Neural Network | PHB | 0.63 (0.43–0.82) | 0.88 (0.71–1.00) | 0.73 (0.55–0.87) |
| XGBoost | PET | 0.95 (0.88–1.00) | 0.76 (0.63–0.87) | 0.84 (0.76–0.92) |
| XGBoost | PHB | 1.00 (1.00–1.00) | 0.64 (0.41–0.86) | 0.78 (0.58–0.92) |

was overall more conservative in its identification, as reflected by lower recall (0.76 and 0.64 for PET and PHB) than Neural Network (1 and 0.88 for PET and PHB) (paired t-test, p-value < 0.01, t = 59.34).

The performance differences between the two models largely stem from the class imbalance in the training data, with many more true negatives than true positives. This led XGBoost to adopt a conservative prediction strategy, yielding high precision by minimizing false positives, which is useful when high-confidence predictions are desired. In contrast, the neural network achieved higher recall by detecting a broader range of potential degraders, though with more false positives. Both models are available in PlasticEnz to accommodate different research goals: XGBoost as the default high-confidence predictions and a more sensitive neural network (via the –sensitive flat) for broader candidate detection.

Different machine learning models are represented by distinct colors, and error bars indicate the 95% confidence intervals.

### PlasticEnz tool workflow description, runtime, and performance

*PlasticEnz* identifies plastic-degrading enzymes from metagenomic data using a two-step search and optional machine learning classification. The tool accepts contigs, genomes, or protein sequences as input and screens them against a curated database using HMMER [25] and DIAMOND [20]. Users can specify the target polymer(s) with the --polymer flag. For PET and PHB, predictions can be refined using a machine learning classifier, namely XGBoost (default) or a more sensitive neural network (activated via --sensitive). If paired metagenomic reads or BAM files are supplied, the tool also estimates gene abundances and reports their raw as well as CPM, RPKM and TPM normalized counts. The output is a report containing information about the predicted plastic-degrading enzyme including its protein sequence, normalized abundance values, sequence similarity scores (E-value, bitscore) and ML prediction score (if applicable). The full workflow is depicted and described in **Fig 2**.

During runtime tests, the tool performed well across inputs of various types and sizes (S5 Table). For smaller inputs such as 1.2Mb single genomes or 100Mb protein files, the real runtime remained under 30 seconds. For medium-sized datasets, including genomes and proteins within 600–650Mb range, *PlasticEnz* workflow completed in under 4 minutes. The runtime for large datasets of nearly 1Gb remained practical, under 5 minutes for proteins and 35 minutes for contigs.

PlasticEnz matches or outperforms existing tools in PHB and PET enzyme annotation on the independent validation set Benchmarking results against are summarized in **Table 3**.

For PET-degrading enzymes, PlasticEnz in Sensitive mode achieved the best overall performance across all metrics (precision = 0.98, recall = 0.96, F1 = 0.97, MCC = 0.95). PlasticEnz in Default mode performed comparably to eggNOG-mapper (F1 = 0.86 vs 0.90; MCC = 0.82 vs 0.87), whereas KEGG-based tools (KOfamKOALA and BlastKOALA) failed to identify any PET hydrolases, resulting in zero values for all metrics.

For PHB-degrading enzymes, all tools showed lower performance, reflecting the higher sequence diversity within this enzyme group. PlasticEnz in Sensitive mode again performed best (F1 = 0.67, MCC = 0.65). The Default mode, eggNOG-mapper, and KOfamKOALA achieved similar scores (all F1 = 0.58, MCC = 0.61), while BlastKOALA performed weakest (F1 = 0.30, MCC = 0.39).

Taken together, these benchmarking results demonstrate that while PlasticEnz in Default mode performs on par with established annotation tools such as eggNOG-mapper, the Sensitive mode consistently outperformed all other annotation tools, most notably for PET-degrading enzymes.

### Application of PlasticEnz to PHB and PET-exposed benthic biofilm communities

We evaluated whether plastizyme candidates detected by PlasticEnz reflected expected environmental gradients and showed sequence similarity to experimentally validated enzymes. Specifically, we compared the abundance and confidence scores of predicted plastizymes (HMM E-value and bitscore, and ML prediction score) across PET, PHB, and control biofilm communities to assess whether putative plastic-degrading enzymes were more prevalent in polymer-exposed

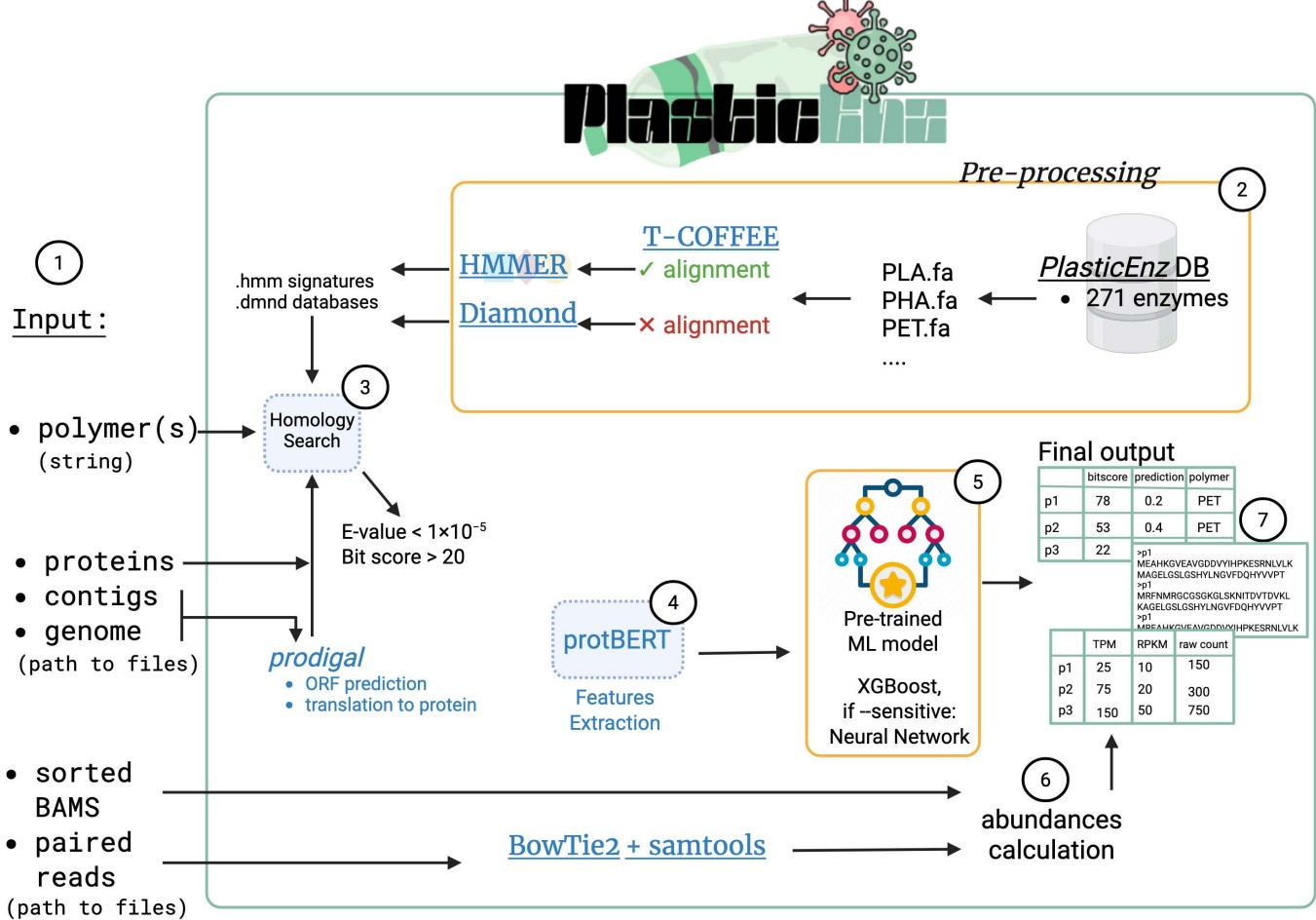

**Fig 2. Overview of the PlasticEnz pipeline.** PlasticEnz identifies candidate plastic-degrading enzymes in metagenomic datasets through a multi-step workflow. (**1**) The user specifies a target plastic polymer or combination of polymers using the --polymer flag and provides a path to one of the following: assembled contigs (--contigs), full genomes (--genome), or protein sequences (--proteins) in FASTA format. Optionally, paired-end sequencing reads or pre-aligned BAM files may be included to quantify gene abundance. (**2**) For nucleotide inputs (contigs or genomes), protein-coding genes are predicted using Prodigal [38] and translated to amino acid sequences. (**3**) The resulting proteins are screened against our custom-made HMM profiles using HMMER [25] or DIAMOND [20] (singleton sequences). Hits must pass default filters (E-value < 1 × 10$^{-5}$, bitscore > 20); HMMER hits are further filtered by bias score (must be < 10% of bitscore). (**4**) Sequences that pass this homology screen are embedded using ProtBERT [39] to generate contextualized feature vectors. (**5**) These embeddings are classified using one of two pre-trained machine learning models: XGBoost (default mode) or a neural network (sensitive mode, activated with --sensitive, optimized for recall). Predictions are returned as probabilities for each supported polymer class (currently PET and PHB). (**6**) If read data is provided, gene abundance is computed either via alignment-based quantification (Bowtie2 [21] + samtools [40]) or directly from sorted BAM files using internal scripts. (**7**) Final outputs include: a summary.csv file listing hits, homology scores, and ML prediction scores; a.fasta file with protein sequences of predicted homologs; and an optional abundance.csv file containing raw and normalized counts (RPKM, TPM, CPM). External tools and packages are marked in blue. This figure was created using BioRender and we have obtained full permission for its use in the publication.

than in control biofilms, as observed in the original study. We further examined the functional relatedness of these predictions by comparing the top-scoring candidates with known plastizymes from the PlasticEnz database using phylogenetic trees and pairwise evolutionary distance analyses. For this evaluation, we applied PlasticEnz in both the default (XGBoost) and sensitive (Neural Network) modes to assembled contigs from PET, PHB, and ceramic biofilm communities, as well as seawater samples (H$_2$O), targeting PET and PHB as polymer variables.

**Table 3. Comparative performance metrics of PlasticEnz vs reference tools annotations.**

| Model / Tool | Polymer | Precision | Recall | F1 | MCC |
|---|---|---|---|---|---|
| PlasticEnz: Default (XGBoost) | PET | 1.00 | 0.76 | 0.86 | 0.82 |
| PlasticEnz: Sensitive (Neural Network) | PET | 0.98 | 0.96 | 0.97 | 0.95 |
| EggNOG | PET | 1.00 | 0.82 | 0.90 | 0.87 |
| KOfamKOALA | PET | 0.00 | 0.00 | 0.00 | 0.00 |
| BlastKOALA | PET | 0.00 | 0.00 | 0.00 | 0.00 |
| PlasticEnz: Sensitive (Neural Network) | PHB | 0.90 | 0.53 | 0.67 | 0.65 |
| PlasticEnz: Default (XGBoost) | PHB | 1.00 | 0.41 | 0.58 | 0.61 |
| EggNOG | PHB | 1.00 | 0.41 | 0.58 | 0.61 |
| KOfamKOALA | PHB | 1.00 | 0.41 | 0.58 | 0.61 |
| BlastKOALA | PHB | 1.00 | 0.18 | 0.30 | 0.39 |

First, we compared HMMER bitscores across all biofilm types. *PlasticEnz* identified 7 putative PETases in the PET biofilm community, with an average HMMER bitscore of 43.9 (SD: 9.6) (**S4 Table**). In comparison, the average HMMER bitscores for seawater and ceramic samples were 52.0 (SD: 12.8) and 41.7 (SD: 12.1), respectively. Next, we examined ML classifier prediction scores under both model settings. As expected, the sensitive model produced much higher average prediction values than the default model. For PET, seawater, and ceramic samples, average scores under the default model were 0.06 (SD: 0.05), 0.05 (SD: 0.04), and 0.04 (SD: 0.06), respectively, compared to 0.6 (SD: 0.2), 0.43 (SD: 0.2), and 0.3 (SD: 0.2) in the sensitive mode (**Fig 3A**). Finally, we quantified the number of proteins with high-confidence predictions, defined as the number of proteins reaching prediction scores > 0.7. In the default mode, no PETases in any sample met this threshold. However, in the sensitive mode, 3 PETases in the seawater sample (1.48 hits per million proteins) and 2 in the PET biofilm sample (1.45 hits per million proteins) passed this cutoff (**Fig 3B**). No high-confidence putative PETases were detected in the ceramic biofilms under either setting.

In the PHB biofilm community, *PlasticEnz* identified 826 putative PHB depolymerases with a high average HMM bitscore of 108.4 (SD: 91.1). In contrast, average HMM bitscores were substantially lower for the seawater (56.7, SD: 29.1) and ceramic communities (60.0, SD: 36.5) (**S4 Table**).

Classifier prediction scores for PHB homologs were higher than those observed for PET, across all modes. In the default (XGBoost) mode, average prediction scores were 0.2 (SD: 0.3) for the PHB biofilm, 0.07 (SD: 0.1) for seawater, and 0.09 (SD: 0.2) for ceramic samples (**Fig 3C**). Again, these values increased substantially under the sensitive (Neural Network) mode, reaching 0.6 (SD: 0.2) for PHB, and 0.5 (SD: 0.3) for both seawater and ceramic samples (**Fig 3C**).

Similarly to PETases, quantified the number of proteins with high-confidence predictions (score > 0.7). Under the default model, 64 proteins in the PHB biofilm (38.5 hits per million proteins), 3 in seawater (1.48 hits per million), and 5 in ceramic biofilms (3.69 hits per million) surpassed this threshold. In contrast, the sensitive mode yielded more high-scoring predictions: 363 in the PHB biofilm (219 hits per million), 61 in seawater (30.1 hits per million), and 45 in ceramic samples (33.1 hits per million) (**Fig 3D**).

To assess the sequence similarity between PlasticEnz-predicted PETases/ PHBases and experimentally validated plastizymes, we compared the top 10 predictions from each model (PHB-default, PHB-sensitive, and PET-sensitive) with sequences from the PlasticEnz database.

For the PHB-default model (**Fig 4A**), most predicted proteins showed close sequence alignment to several known enzymes, with an average evolutionary distance of 2.84 (range: 0.80–6.66). The closest matches included known PHB depolymerases from *Pseudomonas lemoignei* (PHB_SEED_3), *Alcaligenes faecalis* (PHB_SEED_4), and *Ralstonia pickettii* (PHB_SEED_5, PHB_SEED_26), all with distances below 1.6. The most dissimilar hits were to enzymes from *Cupriavidus necator* (e.g., PHB_SEED_41, _44, _36), with distances exceeding 4.5.

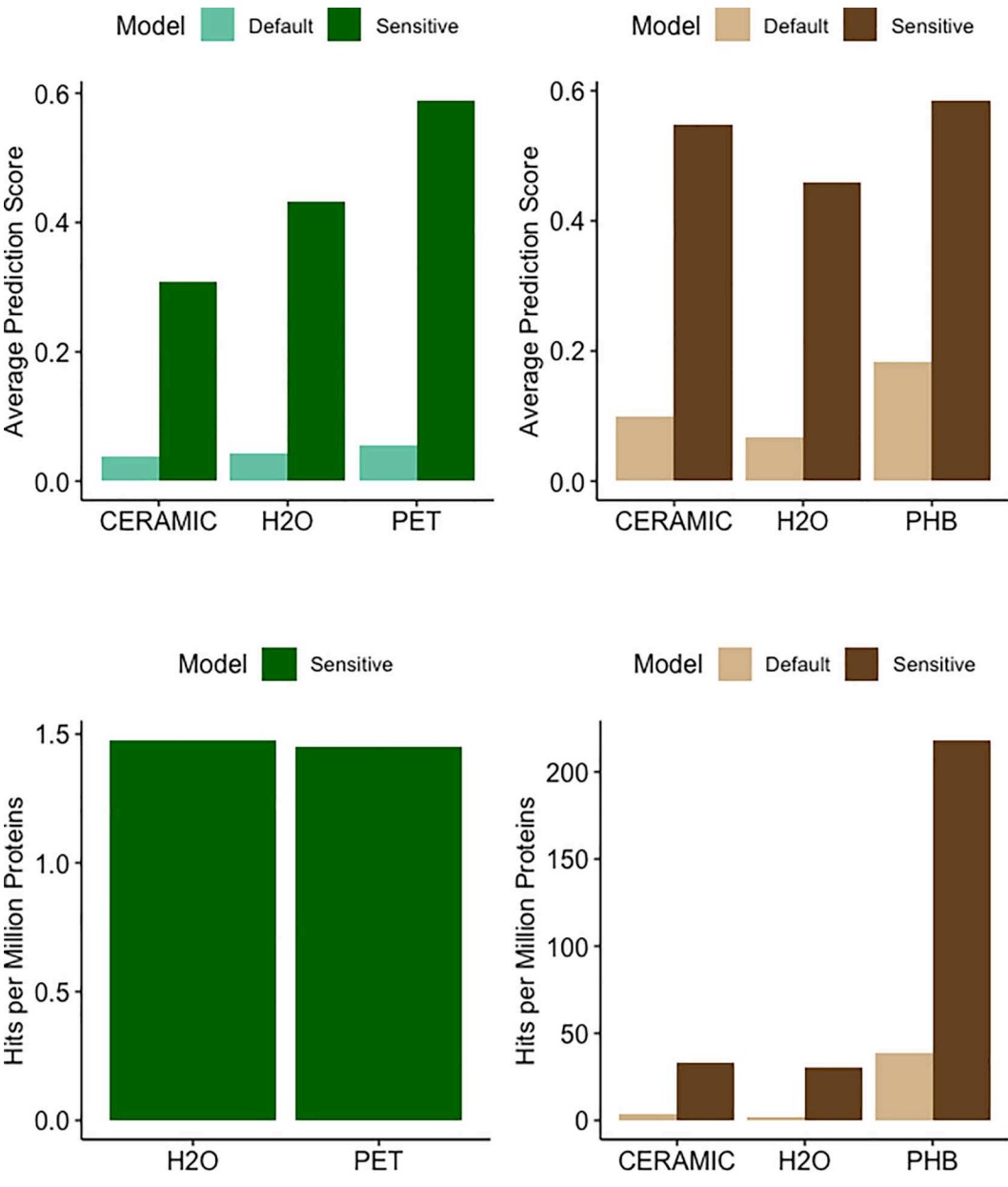

**Fig 3. PlasticEnz predictions of PET- and PHB-degrading enzymes with prediction scores above 0.7 across samples from the Laguna Madre dataset (CERAMIC, PET, PHB, H₂O), shown for both the default and sensitive PlasticEnz models.** Average prediction scores for PET- (**A**, Green) and PHB- (**C**, Brown) degrading enzyme candidates in microbial communities from PET, PHB, ceramic, and seawater (H₂O) samples. Results from the default PlasticEnz model are shown in lighter colours, and results from the sensitive model are shown in darker colours. Total abundances of PlasticEnz-predicted putative enzymes with above 0.7 prediction threshold for PET (**B**, Green) and PHB (**D**, Brown), normalized for sample depth and expressed as proteins per million.

In the PHB-sensitive model (**Fig 4B**), predicted homologs were more diverse, with a higher average evolutionary distance of 3.78 (range: 0.51–10.00), indicating broader but less conserved matches. However, some sequences still aligned well with known depolymerases, including those from *Ralstonia pickettii* (PHB_SEED_5, _26), *Burkholderia cepacia* (PHB_SEED_18), and *Pseudomonas lemoignei* (PHB_SEED_24). The most distant predictions again mapped to

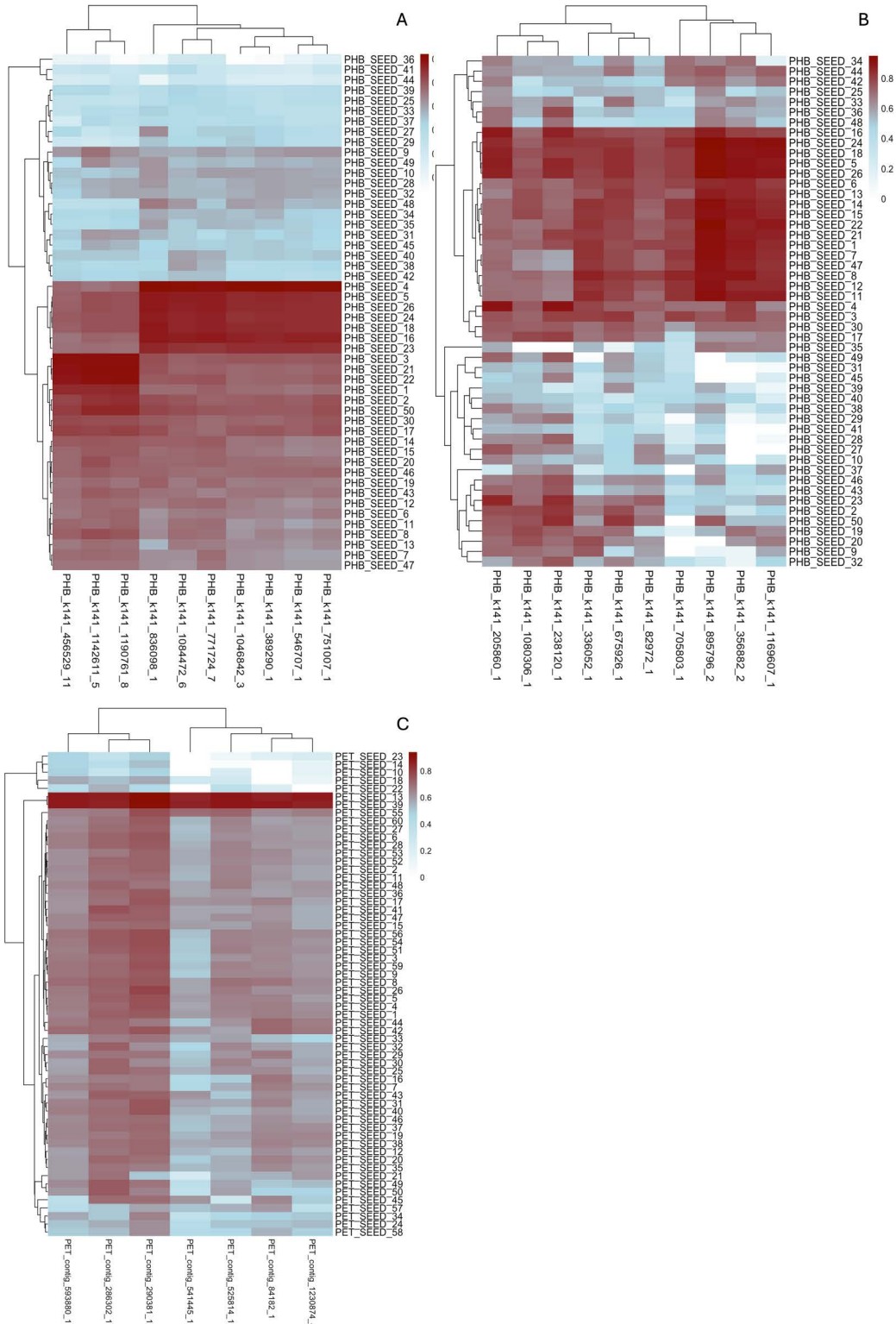

**Fig 4. Protein sequence similarities between the top PlasticEnz predictions in the Laguna Madre dataset and their closest database reference sequences.** Heatmaps display the sequence similarity between the top 10 highest-scoring predicted PETases and PHBases identified by *PlasticEnz* for PHB under default model (**A**), PHB under sensitive model (**B**) and PET under sensitive mode (**C**). *PlasticEnz*-identified plastizymes were compared

against their respective non-redundant reference enzymes from the database of confirmed plastic degrading enzymes. Pairwise evolutionary distances were computed using the LG substitution model and converted into similarity scores ranging from 0 (low similarity) to 1 (high similarity).

*Cupriavidus necator* (PHB_SEED_41, _45, _39) and *Paracoccus denitrificans* (PHB_SEED_31), with distances above 6.3. No PETases were found under the default model, meanwhile the PET-sensitive model (**Fig 4C**) yielded the least conserved matches overall, with an average distance of 3.99 (range: 0.58–10.00). While most predictions showed weak similarity, a few aligned moderately well with reference enzymes such as PET_SEED_13 (*Bacillus subtilis*, p-nitrobenzylesterase), PET_SEED_39 (uncultured bacterium), and PET_SEED_7 (*Pseudomonas aestusnigri*), all showing average distances around 1.2–1.4. The most distant matches were against the fungal PETases from *Fusarium oxysporum* (PET_SEED_10, _14), *Fusarium solani* (PET_SEED_23), and *Humicola insolens* (PET_SEED_22), with distances exceeding 7.4. Furthermore, we compared averaged HMM bit scores and neural network prediction scores for contigs with the highest number of sequence matches to known plastizymes (e.g., *contig_593880_1*, *contig_290381_1*, *contig_286302_1*) and those with the fewest (e.g., *contig_1230874_4*, *contig_541445_1*, *contig_525814_1*). The difference in average HMM scores between the two groups was 48.7 vs. 41.2, meanwhile the ML model prediction scores showed 0.71 vs. 0.50.

### Application of PlasticEnz to microbial communities from plastic-rich and pristine field samples

We applied PlasticEnz to metagenomes from plastic-contaminated soils and pristine hot springs to evaluate whether predicted plastizyme distributions corresponded with environmental exposure gradients. Specifically, we used *PlasticEnz* to identify putative plastic-degrading enzymes in metagenomic samples from plastic-contaminated urban soils (Varamin, Sewapura, Ghazipur) and compared the results to those from thermophilic sediment samples collected from pristine hot springs (Hot_springs_1, _2, _3).

We first examined the average HMM bitscores of PlasticEnz-identified plastizyme candidates across all polymers: P3HP, PBAT, PBS, PBSA, PCL, PEA, PES, PET, PHA, PHB, PHBV and PLA (**Fig 5A**). Plastizyme candidates from plastic contaminated soil samples consistently exhibited higher average HMMER bitscore values across all polymers. In contrast, candidates from hot spring samples had consistently lower HMMER bitscores, and for several polymers (e.g., P3HP, PEA in Hot_spring_3; PES, PET in all samples; PHB in Hot_spring_1) no candidates were detected at all.

Next, we quantified high-scoring plastizymes defined as candidates that exceeded HMMER bitscores of 100, 80, and 50, representing high, medium, and low sequence similarity to the *PlasticEnz* HMM motifs, respectively (**Fig 6**). Across all thresholds and polymer types, *PlasticEnz* consistently detected more high scoring plastizyme candidates in plastic-contaminated soils as opposed to hot spring samples. After normalizing for sequencing depth, Sewapura had the highest number of candidates, followed by Ghazipur and Varamin. As expected, relaxing the HMMER bitscore threshold increased the number of candidates. For example, the number of predicted PLA-degrading enzymes in hot spring samples rose from 3 (bitscore > 100), to 23 (bitscore > 80), and 99 (bitscore > 50). At the lowest threshold (bitscore > 50), the total number of predicted depolymerases also increased across other polymer classes in the pristine hot springs, reaching 4 for PBAT, 47 for PBSA, 14 for PCL, 4 for PHBV, and 46 for PLA.

Next, we examined the *PlasticEnz* ML prediction module by quantifying the number of plasizyme candidates classified as high-confidence PET or PHB depolymerases (prediction score > 0.7) under the default model (**Fig 5B**). As the highest prediction score for proteins in the hot spring samples was 0.0001, only plastic-contaminated sites were considered. As expected, across all contaminated sites, PHB depolymerases were more abundant than PETases. Sewapura exhibited the highest number of high confidence candidates, with 7.28 and 2.43 classified depolymerases per million proteins for PHB and PET, respectively. Varamin followed with 6.23 and 0.78 for PHB and PET while Ghazipuer showed the lowest abundance of PHB depolymerases (5.16 classified depolymerases per million proteins) and no PETases exceeding the 0.7 prediction threshold.

 

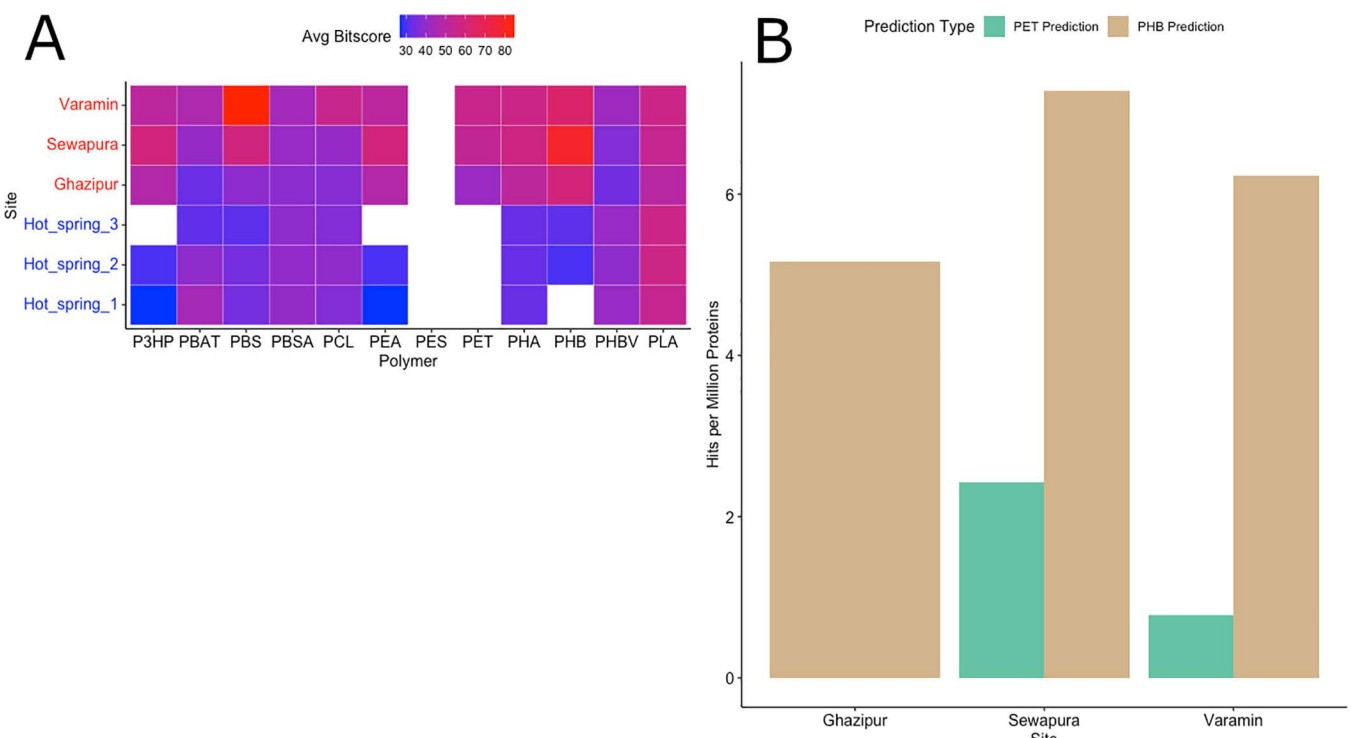

**Fig 5. *PlasticEnz* prediction results for plastic-contaminated urban soil (Ghazipur, Sewapura, Varamin) and pristine hot springs samples (Hot_spring_1–3). (A)** Heatmap showing average HMMER bitscores for PlasticEnz predicted plastic-degrading enzymes across sites and across polymers screened. Sites from pristine hot springs are in blue and plastic contaminated soil samples in red. **(B)** Abundances of proteins (expressed as hits per million proteins) predicted by *PlasticEnz* ML component (default mode) as putative PET (brown) or PHB (green) depolymerases at 0.7 prediction threshold.

Finally, we assessed the sequence similarity between *PlasticEnz*-predicted PETases from contaminated sites and experimentally validated PET-degrading enzymes from the *PlasticEnz* reference database (**Fig 7**). Here, we compared two subsets: (i) PETases predicted with high confidence by the default machine learning component (prediction score > 0.7; **Fig 7A**), and (ii) PETases identified using the stringent HMM bitscore threshold (> 80; **Fig 7B**).

The ML-predicted candidates from Varamin, Sewapura, and Ghazipur clustered closely with multiple known PETases on the PCoA plot, exhibiting higher sequence similarity (PERMANOVA on the clusters, $p > 0.05$, $R^2 = 0.020$; beta-dispersion < 0.05) (**Fig 7A** **and** **7B**). The ML-predicted candidates displayed strong similarity with diverse PETases from the database, including PET_SEED_51 (*Nocardioidaceae*), PET_SEED_60 (*Marinactinospora thermotolerans*), PET_SEED_53 (*Saccharopolyspora flava*), PET_SEED_47 (*unknown bacterium*), and PET_SEED_48 (*Micromonosporaceae*) (**Fig 7C**). The mean pairwise distance between the predicted candidates and known plastizymes was $0.56 \pm 0.12$. In contrast, candidates with high HMMER bitscores formed a more distinct clade, with stronger homology to only two known PETases: PET_SEED_13 (p-nitrobenzylesterase from *Bacillus subtilis*) and PET_SEED_39 (unknown PET hydrolase from an uncultured bacterium) (**Fig 7D**). These sequences were more divergent from known PETases, as reflected by a higher mean pairwise distance ($0.74 \pm 0.04$), and clear separation in the PCoA space (**Fig 7B**).

## Discussion

Our study introduces *PlasticEnz*, a new bioinformatics tool designed to aid researchers in screening for potential plastic-degrading enzymes in complex metagenomic datasets. At the core of *PlasticEnz* is a carefully curated database containing experimentally validated plastic polymer degradation enzymes. This database allowed us to generate the

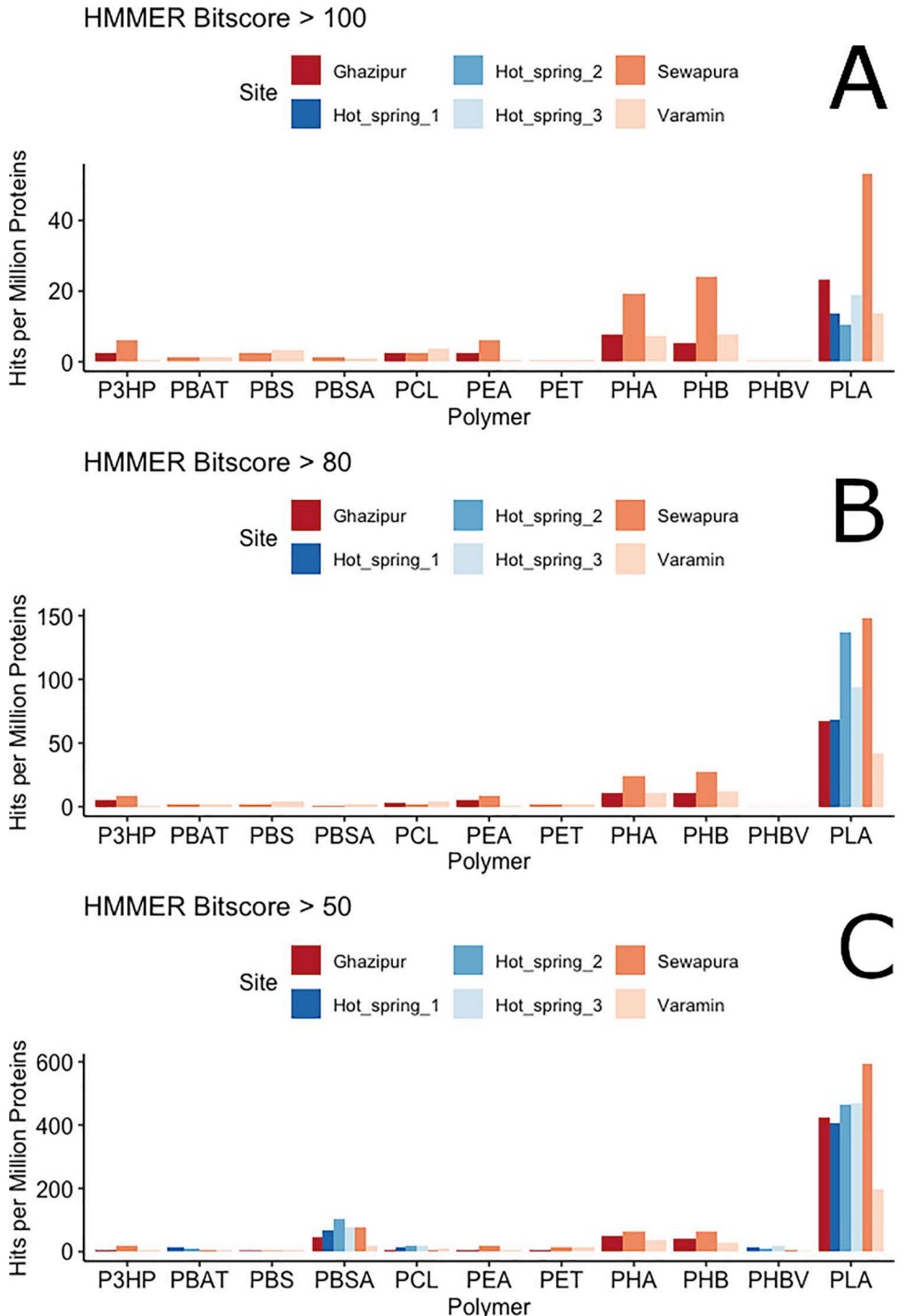

**Fig 6. *PlasticEnz* prediction results for bacteria from plastic-contaminated urban soils (Ghazipur, Sewapura, Varamin) and pristine hot springs samples (Hot_spring_1–3) (C–E)** Bar plots showing the abundances of predicted plastic-degrading enzymes (expressed as hits per million proteins) for each screened polymer (X-axis) and site (colour) across varying HMM bitscore thresholds: > 100 (C), > 80 (D), and >50 (E).

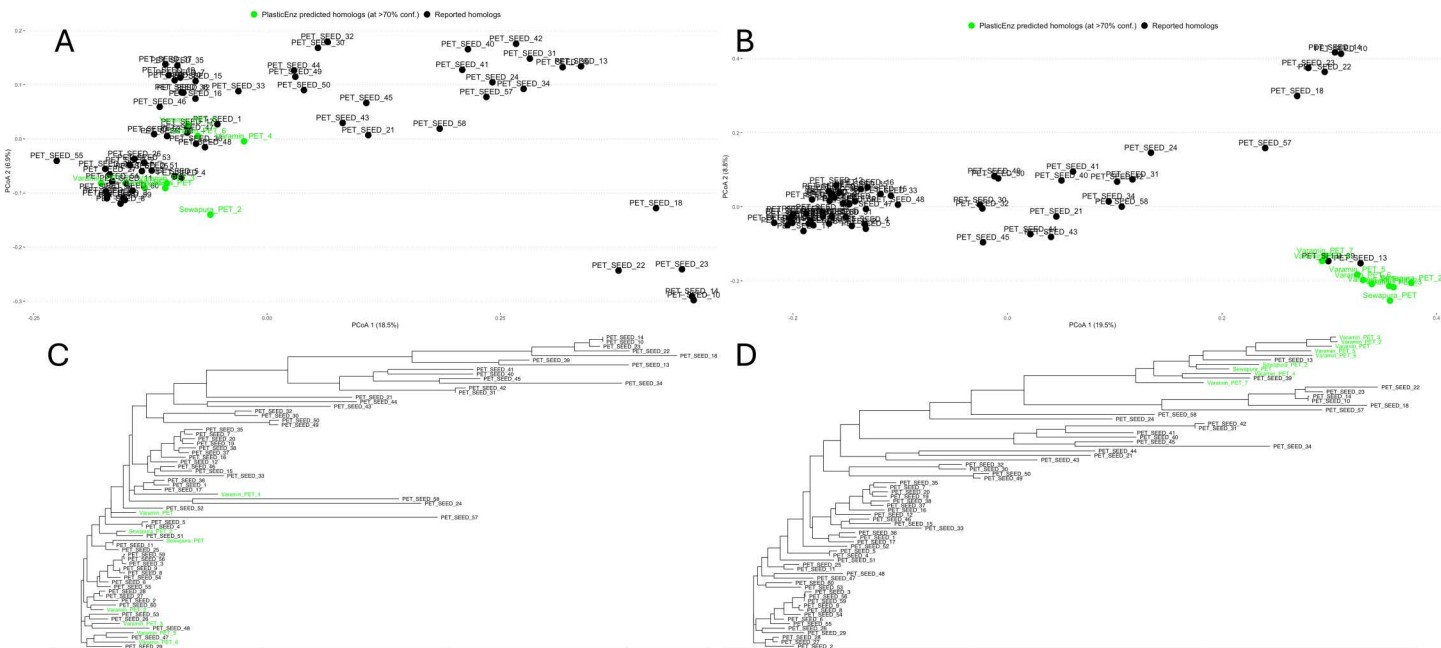

**Fig 7. Evolutionary relatedness of *PlasticEnz* predicted PETases to known reference PET-degrading enzymes. (A, B)** Principal Coordinates Analysis (PCoA) based on Fitch distances showing sequence-level similarities between *PlasticEnz*-predicted putative PETases (green) and known reference PETases (black) under high-confidence ML predictions (≥ 0.7) (**A**) or HMMER bitscore above 80 (**B**). **(C, D)** Corresponding phylogenetic trees derived from Clustal Omega alignments and built with FastTree under the LG substitution model, showing clustering of predicted PETases (green) with reference enzymes (black).

polymer-specific Hidden Markov Models (HMM) and provided a resource for training a machine learning (ML) classifier for PETases and PHBases annotation. Benchmarking against common protein annotation tools shows that PlasticEnz's strategy of combining custom and polymer-specific HMM motifs with pre-trained ML component outperforms common tools like KEGG mapper or eggNOG-mapper in annotation of PETases and PHBases.

A critical step in developing the machine learning component of PlasticEnz was the construction of clearly defined positive and negative datasets. For the positive set, i.e., ground truth sequences shown to degrade plastic polymers experimentally, we used the data from PlasticEnz and PlasticDB [18] databases. However, creating a reliable negative, defined as a set of sequences of distantly homologous proteins without proven ability to degrade plastic polymers was challenging due to the widespread presence of plastic contamination across all the planet's environments [41–43]. To overcome this, we used sequences from well-characterized bacteria that depend on hosts or live parasitically, making them unlikely to synthesise extracellular enzymes targeting plastic polymers. Furthermore, given the limited size of our dataset, we avoided deep learning architectures, which are prone to overfitting under data-scarce conditions [44,45]. Instead, we employed simpler, more interpretable models such as decision trees, gradient-boosted trees (XGBoost), and a shallow neural network (single hidden layer). To further reduce overfitting risk and improve generalizability, we incorporated regularization techniques such as early stopping and dropout for the neural network [46,47] and hyperparameter optimization for all models. In line with previous findings [37], which evaluated thirteen classifiers on plastic degradation enzyme classification problems, XGBoost emerged as the most suitable model. Overall, all models showed strong performance for two polymers: poly(ethylene terephthalate) (PET) and polyhydroxybutyrate (PHB), but consistently underperformed on others like PLA, PBAT, PBSA, and PCL. For these underrepresented classes, precision, recall, and $F_1$ scores were consistently low and exhibited wide confidence intervals. This is a common effect of class imbalance, where the number of

true positive examples is greatly outweighed by negative instances. As the size of the positive set increases, performance metrics such as precision and recall improve accordingly, which is supported by a positive correlation coefficient (Pearson, $r = 0.81$) between the $F_1$ score and the number of sequences in the positive set. These results highlight a key limitation in training ML models for plastic degradation classification: the limited number of ground truth sequences. As more enzymes are identified and incorporated into the training set, model performance is expected to improve accordingly. Therefore, to ensure reliable predictions, the *PlasticEnz* prediction module is limited to PET and PHB, the only polymers with sufficient training data. Users can choose between a default XGBoost model optimized for precision, or a more sensitive neural network model that emphasizes recall, allowing flexibility based on research priorities.

To showcase *PlasticEnz* capabilities, we applied it to two distinct experimental setups: (1) a controlled microcosm study conducted in a hypersaline lagoon (Laguna Madre), and (2) field samples from plastic-contaminated and pristine environments. In the Laguna Madre dataset, originally analyzed by Pinell et al., 2022 [48] metagenomic analysis focused on biofilm communities growing on PET, PHA, and ceramic substrates over 28 days. Consistent with the original findings, *PlasticEnz* did not detect significant enrichment of PET-degrading enzymes in PET biofilms compared to seawater or ceramic controls. No high-confidence PETase was found in any sample under the default mode, and even under the sensitive mode, PET biofilms did not show increased PETase predictions relative to seawater. This aligns with Pinell's interpretation that the remote location and low plastic exposure likely limited the selection pressure for PET-degrading enzymes in these communities. In contrast, the original study reported that communities extracted from PHA biofilms were significantly enriched in PHB depolymerases in comparison to control biofilms. Using *PlasticEnz*, we identified over 800 putative PHB depolymerases in these communities, with significantly higher average HMM bitscore and ML prediction scores relative to control samples. Notably, heatmaps showed that many of these predicted enzymes shared high sequence similarity to experimentally validated entries in the SEED database. However, *in-vitro* assays will be required to confirm their functional activity.

To further demonstrate the capabilities of *PlasticEnz*, we applied the tool to screen microbial communities from two contrasting environments. While plastic-contaminated urban soils of Varamin, Sewapura and Ghazipur served as representative examples of polymer-enriched ecosystems, identifying a truly pristine, plastic-free environment proved challenging due to the ubiquity of microplastics [27,28,41–43,49,50]. We selected the geothermal area of the Mutnovsky volcano in Kamchatka, a remote region with minimal anthropogenic activity, as a proxy for a low-contamination environment. This site is characterized by extreme physicochemical conditions, including broad gradients in temperature and pH [51]. Due to limited organic carbon sources, these communities are dominated by obligately or facultatively chemolithoautotrophic bacteria and the presence of hydrolytic enzymes capable of degrading complex plastic polymers was expected to be minimal [52,53]. As expected, plastic-contaminated soils contained putative plastic-degrading homologs with higher average HMM bitscores and greater numbers of depolymerases than pristine hot spring samples. The strongest enrichment was seen in Sewapura, followed by Ghazipur and Varamin. Notably, all sites, including hot springs, showed elevated PLA depolymerase signals. This is because most PLA-degrading enzymes belong to the serine protease family, a group of diverse and taxonomically widespread enzymes commonly found across bacterial, fungal, and archaeal taxa [8,54–56].

Next, we evaluated the *PlasticEnz* default prediction module using a classification threshold of 0.7. None of the putative PET or PHB homologs from the hot spring samples met this threshold; in fact, all prediction scores were below 0.1. Among the plastic-contaminated sites, PETase predictions exceeding the 0.7 confidence threshold were observed in communities from Sewapura and Varamin but not in Ghazipur. Meanwhile, high-confidence predictions of PHB depolymerases were common to all sites. This likely reflects underlying biological differences: PHB depolymerases are widespread due to the role of PHA polymers in microbial carbon and energy storage, whereas PET degradation requires evolved adaptations of esterases or cutinases [7,14]. As a result, PET-degrading enzymes remain rare and are typically associated with long-term plastic exposure.

Lastly, we conducted a comparison study by clustering *PlasticEnz*-identified putative PET homologs to previously reported and functionally validated PETases (SEED sequences). This analysis was split into two comparisons: one

 

including high-scoring homologs (HMM bitscore > 80), and a second including sequences assigned a prediction score greater than 0.7 by the XGBoost classifier (default mode). ML-predicted PETase homologs from plastic-contaminated sites clustered with several known reference PETases, while the high HMM score sequences formed a distinct clade associated with two particular SEED reference enzymes. These results demonstrate that the *PlasticEnz* prediction module can identify candidate PETases with strong evolutionary ties to a broader range of validated enzymes, extending beyond the reach of traditional homology-based methods like HMMs, especially in complex metagenomic datasets. However, it is also possible that the model may be biased toward well-represented sequence patterns in the training data, potentially reducing its ability to detect PETases that were underrepresented in the training set.

While *PlasticEnz* offers a streamlined and accessible framework for identifying putative plastic-degrading enzymes, several limitations should be considered. First, the presented analysis focused on high-confidence predictions based on stringent HMM bitscore and ML probability thresholds; users are encouraged to apply similar cutoffs to ensure reproducibility. Additionally, the speed of the pipeline scales with dataset size, and RAM requirements increase accordingly, particularly during ProtBERT embeddings and contigs translation to proteins. To address this, a --gpu option is provided to accelerate tokenization, and the tool allows the use of multiple cpu cores via the --cores flag. However, since prodigal is not optimised to execute on multiple cores, the users are advised to adjust computational resources as needed for bigger datasets (over 1Gb). Importantly, *PlasticEnz* relies on sequence homology, and its predictions remain putative until validated through *in vitro* assays. The accuracy and breadth of the tool are ultimately constrained by the number and diversity of experimentally verified plastic-degrading enzymes available for model training. As the field advances and more polymer-degrading enzymes are experimentally validated, *PlasticEnz* will be continuously updated with expanded HMM profiles and retrained machine learning classifiers, enhancing its predictive accuracy, polymer coverage, and utility for metagenomic screening in diverse environments.

## Materials and methods

### Curation and processing of plastic-degrading enzyme sequence data

We systematically collected data on experimentally confirmed plastic degradation enzymes, including their protein sequences, host organisms, reaction details (including catalysts and conditions), references to relevant publications, and cross-references to external databases such as UniProt [57], NCBI [58], and KEGG [59]. This comprehensive information was extracted from published literature and stored in a *PlasticEnz* SQL database (available within the *PlasticEnz* package). To enhance our dataset, we also incorporated data from PlasticDB [18], a publicly available database specializing in plastic degradation enzymes. In total, the two combined databases resulted in 422 protein sequences of diverse plastic-degrading enzymes. These protein sequences were then pooled together into combined fasta files based on their respective polymer substrate. To eliminate redundancy caused by the database merge, we clustered each combined fasta at 95% similarity using CD-HIT2 [60]. Following clustering, multiple sequence alignments (MSA) were performed for each grouped set of unique proteins using T-Coffee in expresso mode [61], which integrates protein structure. The resulting alignments were refined to maintain only those with average to good alignment score (TCoffee alignment score > 50). Sequences that failed to align adequately were pressed into the DIAMOND database [20] (S2 Table). MSAs were used to generate HMM profiles using the built-in *hmmbuild* function from the HMMER suite [25] (S2 Table).

### Preparation of training and test data sets

To build the negative dataset, we used our previously generated HMM profiles to identify homologous sequences in bacterial genomes that are unlikely to be involved in plastic degradation activity. Specifically, we focused on representative Refseq genomes from well-studied host-dependent or parasitic bacteria (e.g., *Escherichia coli*, *Chlamydia trachomatis*, *Staphylococcus aureus*) (S3 Table). The resulting distant homologous sequences formed the negative dataset (410 sequences). For the positive dataset, we used our previous set of experimentally confirmed bacterial plastic-degrading

enzymes. Positive and negative datasets were combined and clustered at 95% identity with CD-HIT2 [60] to obtain 502 unique clusters. To avoid the presence of highly similar sequences between training and test datasets, we randomly split entire clusters into either the training set (80%, 606 protein sequences) or the test set (20%, 200 protein sequences). The feature matrix used for the model training contained one-hot encoded annotations indicating the specific plastic polymer(s) degraded by each enzyme.

### Embeddings generation with ProtBERT

To generate protein embeddings, we used ProtBERT [39], a pre-trained transformer-based model specifically trained on protein sequences. ProtBERT and its tokenizer were loaded from the Hugging Face Transformers library. Sequences were preprocessed by trimming ambiguous residues ('X', 'x') from sequence ends, verifying that only standard amino acids (ACDEFGHIKLMNPQRSTVWY) were present, and formatting each sequence by inserting spaces between individual residues. ProtBERT generated contextual embeddings for each amino acid, which were then aggregated into a fixed-length embedding vector for each protein by mean pooling across the sequence length.

### Machine learning model selection and evaluation

Full training procedures, hyperparameter settings, and performance metrics are detailed under '*Machine Learning Model Selection and Evaluation*' in S2 Data.

   We evaluated three classifiers: neural network, Random Forest, and XGBoost, using precomputed protein embeddings and one-hot encoded features. All models were trained on a multi-label dataset for PET and PHB degradation using an 80/20 train/test split. Hyperparameters were optimized via grid search and validated on held-out data. Final models were retrained on the full training set and evaluated on the independent test set using precision, recall, and F1-score (S2 File). Model performance (F1-score, precision and recall) was compared between Neural Network and XGBoost classifiers using paired *t*-tests. Normality of metric distributions was confirmed using Shapiro–Wilk tests ($p > 0.05$). Mean, standard deviation, test statistics, and p-values were reported for each metric. Models were implemented and trained with PyTorch, scikit-learn, and XGBoost libraries. Prediction scores from XGBoost and neural network models are included in the PlasticEnz module and represent per-class probabilities for each polymer.

### Benchmarking PlasticEnz

To evaluate the performance of PlasticEnz against existing protein tools, we benchmarked PlasticEnz in default (XGBoost) and sensitive (Neural Network) modes against widely used functional annotation platforms: EggNog-mapper (v. 2.1.12)[73], KOfamKOALA[74] and BlastcKOALA[75] (S1 Data). Since no public, gold-standard metagenomics dataset containing experimentally validated plastizymes and reliable negative examples currently exists, we conducted the benchmarking evaluation on the independent validation dataset (S2 and S3 and S4 Data files). This dataset comprises of protein sequences that were excluded from model training but utilized for classifier performance assessment in **Fig 1** & **Table 2**. In brief, proteins associated with PET and PHB degradation were extracted from the independent validation set and annotated using each platform under their default parameters. A correct annotation for PET-degrading enzymes was defined as assignment to K21104 (polyethylene terephthalate hydrolase), while correct annotations for PHB-degrading enzymes corresponded to K05973 or K03932 (poly(3-hydroxybutyrate) depolymerase and polyhydroxybutyrate depolymerase, respectively) and were classified as true positives. Any alternative KO assignment or absence of assignment for this set was considered a false negative. The negative control set consisted of proteins containing homologous polyester-degrading motifs but derived from bacteria known to lack plastic-degrading capability (e.g., parasitic or host-dependent organisms). If any of these proteins were incorrectly annotated as K21104, K05973, or K03932, they were classified as false positives; otherwise, they were considered true negatives. Following the same criteria established as during the bootstrap-based performance estimation, predictions with a PlasticEnz score

exceeding 0.5 for PET or PHB were classified as positive hits. Proteins with scores below 0.5 or without any annotation were considered negative (no hit).

### Evaluation sets

A full description of datasets, processing steps, and analysis parameters is provided under '*Evaluation sets*' in S1 File.

In brief, we analyzed paired-end Illumina metagenomes from a 28-day microcosm study by Pinnell et al. (2019) [NCBI: PRJEB15404] [48], which examined biofilm communities developed on PET, PHA, and ceramic beads in a hypersaline lagoon. Four sample groups were used: PET, PHA (PHB), ceramic, and seawater controls ($H_2O$). Raw reads were quality-filtered with Fastp [62] (default settings) and assembled using MEGAHIT (default settings) [63]. Assembled contigs were used directly as an input for *PlasticEnz*. Hit counts for PHB/PET depolymerases were normalized per million predicted proteins for comparison across samples. To assess functional similarity, top 10 enzyme predictions from PET-sensitive, PHB-default, and PHB-sensitive models were compared to *PlasticEnz*'s curated SEED reference set (CD-HIT, 90% identity). Sequences were aligned with *MUSCLE* [64], trimmed using *trimAl* (–automated1 setting) [65], and evolutionary distances were calculated with the LG model [66] in *phangorn* (R). Normalized pairwise distances were visualized as similarity heatmaps.

We validated *PlasticEnz* custom HMM motifs using datasets from environments with contrasting plastic exposure levels. Thermophilic hot spring sediment samples from Kamchatka, Russia (NCBI: PRJNA419239), served as a low-exposure control, while plastic-contaminated soil microbiomes from India and Iran, including Sewapura (NCBI: PRJNA1077790) [67], Varamin (NCBI: PRJNA924045) [68], and Ghazipur (NCBI: PRJNA388130), served as high-exposure test sites. Downloaded raw paired-end reads were processed identically to Laguna madre samples. High-confidence PETase predictions (prediction score > 0.7 (default model) or HMMER bitscore > 80) were aligned with SEED references using *Clustal Omega* [69]. Sequence similarity was assessed with Fitch distances [70] (*seqinr*) [71] and visualized using Principal Coordinates Analysis (*ape*) [72]. Phylogenies were generated with *FastTree* [73] using trimmed alignments (*trimAl* [65], gap threshold 0.5).

Analyses were performed in R (v4.2) and Python (v3.11.11). Visualizations were generated using *ggplot2*, *ggpubr*, *patchwork* and *pheatmap*.

### Code availability and supporting information

*PlasticEnz* is freely available at https://github.com/msysbio/PlasticEnz

All relevant data underlying the findings of this study, including the raw data used for model training, benchmarking, intermediate files, scripts to generate figures and tables are provided within the manuscript, Supporting Information files and zenodo repository (10.5281/zenodo.15395662). No additional restrictions apply.

### Supporting information

**S1 Table. Supported polymers and available search strategies.**
(XLSX)

**S2 Table. Seed sequence counts for hMMER and DIAMOND searches.**
(XLSX)

**S3 Table. Description of positive and negative training sequences.**
(XLSX)

**S4 Table. PlasticEnz results for Laguna Madre microcosm study.**
(XLSX)

**S5 Table. Speed tests results.**
(XLSX)

**S1 File. Supplementary Materials & Methods.**
(DOCX)

**S2 File. Formulas for Precision, Recall and F1-Score.**
(PDF)

**S1 Data. Benchmarking values calculation.**
(XLSX)

**S2 Data. Protein sequencs used for benchmarking tests of negative set.**
(FASTA)

**S3 Data. Protein sequencs used for benchmarking tests of PET set.**
(FASTA)

**S4 Data. Protein sequencs used for benchmarking tests of PHB set.**
(FASTA)

## Acknowledgments

We thank Maxime Greffe and Hubert Krukowski for kindly testing the tool.

## Author contributions

**Conceptualization:** Anna Krzynowek.

**Funding acquisition:** Anna Krzynowek, Karoline Faust.

**Investigation:** Karoline Faust.

**Methodology:** Anna Krzynowek.

**Project administration:** Anna Krzynowek, Karoline Faust.

**Software:** Anna Krzynowek, Jasper Snoeks.

**Supervision:** Anna Krzynowek, Karoline Faust.

**Validation:** Anna Krzynowek.

**Visualization:** Anna Krzynowek.

**Writing – original draft:** Anna Krzynowek.

**Writing – review & editing:** Anna Krzynowek, Karoline Faust.

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
