## [Decision Letter · Decision Letter 0]

20 Aug 2025

PCOMPBIOL-D-25-01110

PlasticEnz: An integrated database and screening tool combining homology and machine learning to identify plastic-degrading enzymes in meta-omics datasets

PLOS Computational Biology

Dear Dr. Krzynowek,

Thank you for submitting your manuscript to PLOS Computational Biology. As with all papers, your manuscript was reviewed by members of the editorial board. Based on our assessment, we have decided that the work does not meet our criteria for publication and will therefore be rejected. If external reviews were secured, reviewers' comments will be included at the bottom of this email.

We are sorry that we cannot be more positive on this occasion. We very much appreciate your wish to present your work in one of PLOS's Open Access publications. Thank you for your support, and we hope that you will consider PLOS Computational Biology for other submissions in the future.

Yours sincerely,

Eduardo Jardón-Valadez

Academic Editor

PLOS Computational Biology

Shihua Zhang

Section Editor

PLOS Computational Biology

**Additional Editor Comments (if provided):**

Dear Dr. Karoline Faust,

We thank you for submitting your manuscript to PLOS Computational Biology and for considering our journal as a venue for your research. After a thorough review process and careful consideration of the reviewers’ comments, we have reached a decision regarding your submission.

Unfortunately, we regret to inform you that we are unable to accept your manuscript for publication in PLOS Computational Biology. While the study presents interesting findings, the reviewers have raised substantial concerns that may not be properly addressed through revision.

We truly appreciate the effort you have dedicated to this work and the opportunity to evaluate your manuscript. We encourage you to consider the reviewers’ feedback, which we believe may be of assistance in further developing your research for submission to another journal.

Thank you again for your interest in publishing with PLOS Computational Biology. We wish you every success with your future research endeavors.

**Reviewers' Comments (if peer reviewed):**

Reviewer's Responses to Questions

**Comments to the Authors:**

Reviewer #1: This manuscript presents the PlasticEnz tool, which integrates homology-based searches with machine learning, and validates its ability to detect plastic-degrading enzymes across diverse environmental datasets. However, the work lacks sufficient comparison with existing tools, and the description of polymer coverage and limitations is not sufficiently clear. The following points require clarification:

1. Some abbreviations (e.g., PEA, P3HP, PBAT) are not defined at their first occurrence in the main text; the full names should be provided.

2. Figure 1 presents only bar charts; it is recommended to include confusion matrices for each polymer in the Supplementary Materials to facilitate interpretation of false positives and false negatives.

3. Although the Neural Network section mentions early stopping and dropout, key hyperparameters (e.g., learning rate, number of hidden layer nodes, dropout rate) are not provided; these should be included in the Supplementary Materials.

4. It is recommended to introduce a concise table or a more systematic paragraph summarizing the characteristics of existing tools (polymer coverage, algorithms, usability, validation type) and to clearly state PlasticEnz’s relative advantages in these aspects.

5. The current results compare only HMM and ML (internal methods) without benchmarking against existing tools or general protein function prediction models.

6. The rationale for selecting the XGBoost model in PlasticEnz appears overly simplistic; a more comprehensive evaluation strategy should be considered.

7. The authors propose an effective strategy. In future work, widely developed deep learning techniques and their application to more complex scenarios, such as cancer gene identification (DOI: 10.1038/s41551-024-01312-5) and RNA m6A modification site prediction (DOI: 10.1038/s42003-025-08265-8 and DOI: 10.1016/j.patcog.2025.111541) could be further explored.

Reviewer #2: the article describes a database and and tool/pipeline to identify plastic-degrading enzymes. Whereas the topic is very relevant and interesting, the article is very difficult to read. In addition, there is no proper benchmarking even comparing the the methods described in the manuscript between them (how do we know that a simple similarity search is not enough?) or with other methods ( why your approach is better than others?). Moreover, most of the article describes the application to several datasets but even if the results make sense the lack of benchmarking prevent their actual application.

To solve this issue, in addition to the creation of the dataset already described in the article the authors must carry out a blind test that resembles an actual application case, e.g. a metagenome where plastic-degrading characterized where similarities with the training set have been reduced to a minima. All methods described have to be tested on this dataset considering annotated plastic-degrading genes as positive examples and all remaining coding genes as negative. At least precision, recall and F1, probably MCC, must be reported for each method and compared. Additional analysis on misclassified genes should be also be reported (do methods always fail on the same examples? or is each method better for certain types?). Please ensure DOME-ML guidelines are followed (https://dome-ml.org/).

There are several minor issues to be fixed but these are minor compared to the above. For example, what is the number of sequences used to generate HMMs, are they enough?

**Have the authors made all data and (if applicable) computational code underlying the findings in their manuscript fully available?**

Reviewer #1: None

Reviewer #2: Yes

PLOS authors have the option to publish the peer review history of their article (what does this mean? ). If published, this will include your full peer review and any attached files.

**Do you want your identity to be public for this peer review?** For information about this choice, including consent withdrawal, please see our Privacy Policy .

Reviewer #1: No

Reviewer #2: **Yes: ** Alberto J.M. Martin

---

## [Decision Letter · Decision Letter 1]

6 Jan 2026

Dear Miss Krzynowek,

We are pleased to inform you that your manuscript 'PlasticEnz: An integrated database and screening tool combining homology and machine learning to identify plastic-degrading enzymes in meta-omics datasets' has been provisionally accepted for publication in PLOS Computational Biology.

Best regards,

Eduardo Jardón-Valadez

Academic Editor

PLOS Computational Biology

Shihua Zhang

Section Editor

PLOS Computational Biology

Dear Dr Anna Maria Krzynowek,

After reviewing your revised manuscript, we found that all reviewers’ concerns have been adequately addressed. The revised version now includes benchmarks comparing available annotation tools on related datasets. I believe that the methods you implemented will contribute to the development of novel technologies for mitigating microplastics in the environment.

I therefore recommend your manuscript for publication in PLOS Computational Biology.

Sincerely,

Eduardo Jardón-Valadez

Reviewer's Responses to Questions

**Comments to the Authors:**

Reviewer #1: All of my concerns have been addressed in this revised manuscript.

**Have the authors made all data and (if applicable) computational code underlying the findings in their manuscript fully available?**

Reviewer #1: None

PLOS authors have the option to publish the peer review history of their article (what does this mean? ). If published, this will include your full peer review and any attached files.

**Do you want your identity to be public for this peer review?** For information about this choice, including consent withdrawal, please see our Privacy Policy .

Reviewer #1: No

---

## [Editor Report · Acceptance letter]

PCOMPBIOL-D-25-01110R1

PlasticEnz: An integrated database and screening tool combining homology and machine learning to identify plastic-degrading enzymes in meta-omics datasets

Dear Dr Krzynowek,

I am pleased to inform you that your manuscript has been formally accepted for publication in PLOS Computational Biology. Your manuscript is now with our production department and you will be notified of the publication date in due course.

With kind regards,

Judit Kozma
